# Nano-metal diborides-supported anode catalyst with strongly coupled TaO$_x$/IrO$_2$ catalytic layer for low-iridium-loading proton exchange membrane electrolyzer

Yuannan Wang[1], Mingcheng Zhang[1], Zhenye Kang [2], Lei Shi[1], Yucheng Shen[1], Boyuan Tian[3], Yongcun Zou[1], Hui Chen [1]✉ & Xiaoxin Zou [1]✉

The sluggish kinetics of oxygen evolution reaction (OER) and high iridium loading in catalyst coated membrane (CCM) are the key challenges for practical proton exchange membrane water electrolyzer (PEMWE). Herein, we demonstrate high-surface-area nano-metal diborides as promising supports of iridium-based OER nanocatalysts for realizing efficient, low-iridium-loading PEMWE. Nano-metal diborides are prepared by a novel disulphide-to-diboride transition route, in which the entropy contribution to the Gibbs free energy by generation of gaseous sulfur-containing products plays a crucial role. The nano-metal diborides, TaB$_2$ in particular, are investigated as the support of IrO$_2$ nanocatalysts, which finally forms a TaO$_x$/IrO$_2$ heterojunction catalytic layer on TaB$_2$ surface. Multiple advantageous properties are achieved simultaneously by the resulting composite material (denoted as IrO$_2$@TaB$_2$), including high electrical conductivity, improved iridium mass activity and enhanced corrosion resistance. As a consequence, the IrO$_2$@TaB$_2$ can be used to fabricate the membrane electrode with a low iridium loading of 0.15 mg cm$^{-2}$, and to give an excellent catalytic performance (3.06 A cm$^{-2}$@2.0 V@80 °C) in PEMWE——the one that is usually inaccessible by unsupported Ir-based nanocatalysts and the vast majority of existing supported Ir-based catalysts at such a low iridium loading.

Proton exchange membrane water electrolyzer (PEMWE) is an advanced hydrogen production technology that can achieve large current density (>1 A cm$^{-2}$), high hydrogen purity (>99.99%), and fast response (<5 s) towards dynamic electricity input[1–3]. The heart of PEMWE is a catalyst coated membrane (CCM) composed of solid electrolyte (i.e., perfluorosulfonic membrane), cathode catalyst layer, and anode catalyst layer (Supplementary Fig. 1). Industrially, Pt and Ir-based noble metals are required to catalyze the hydrogen evolution reaction (HER) at the cathode and the oxygen evolution reaction (OER) at the anode, respectively[4–8]. Due to the scarcity and expensiveness of Pt and Ir, strategies that can decrease noble metal loading in CCM and simultaneously maintain good catalytic performance deserve to be strongly pursued. The successful method is to anchor catalyst nanoparticles on a high-surface-area conductive support. An effective

[1]State Key Laboratory of Inorganic Synthesis and Preparative Chemistry, College of Chemistry, Jilin University, Changchun 130012, China. [2]State Key Laboratory of Marine Resource Utilization in South China Sea, Hainan Provincial Key Lab of Fine Chemistry, School of Chemical Engineering and Technology, Hainan University, Haikou 570228, China. [3]State Key Laboratory of Advanced Transmission Technology (State Grid Smart Grid Research Institute Company Limited), Beijing 102209, China. ✉e-mail: chenhui@jlu.edu.cn; xxzou@jlu.edu.cn

support material can enhance the dispersion of catalyst nanoparticles, stabilize them from agglomeration and thus benefit the exposure of large active surface areas[9–12]. In the approach, Pt/C as HER catalyst is commercially employed to construct a cathode catalyst layer of CCM with a low loading of <0.5 mg$_{Pt}$ cm$^{-2}$ [13–15]. But the carbonous supports are not suitable for Ir-based OER nanocatalysts because of their instability in highly oxidizing environments. Thanks to the lack of suitable support materials, the Ir loading in CCM is usually as high as a few mg cm$^{-2}$ [16–18]. Therefore, it is highly desirable to find non-carbonous support materials possessing good electrical conductivity, large specific surface area, and high resistance from acid corrosion and oxidative decomposition for realizing low-iridium-loading PEMWE.

To this end, we turn our attention to boride ceramics, which are generally characterized by high melting points, high hardness, high chemical resistance, and high electrical conductivity. A number of various borides can be produced and used as, for example, high-temperature structural materials, anti-oxidation coating materials and wear resistant materials in industry[19–25]. Despite their great potential, boride ceramics have not been investigated as the support of Ir-based OER nanocatalysts for constructing anode catalyst layer of CCM in PEM electrolyzer. In part, this is because some potential borides (e.g., TaB$_2$) prepared with some existing methods are usually composed of micrometer-sized particles with low surface areas, and sometimes contain large amounts of defects and impurities[26–30]. Thus, synthesizing high-quality borides with large surface areas is an essential pre-requisite for meeting our goal of making efficient boride-supported anode catalysts and applying them in low-iridium-loading PEMWEs.

Herein, we present the entropy-driven synthesis of high-surface-area nano-metal diboride from the corresponding disulphide, and demonstrate the practical usefulness of boride ceramics, especially TaB$_2$, as the effective support material of anode nanocatalyst IrO$_2$ in PEMWE. The TaB$_2$-supported IrO$_2$ nanocatalyst (denoted as IrO$_2$@TaB$_2$) possesses a large electrical conductivity (0.1–0.2 S cm$^{-1}$) similar to nano-IrO$_2$ itself, shows about 10 times higher iridium mass activity than IrO$_2$, and exhibits much lower iridium leaching than IrO$_2$ during electrocatalysis. When integrated into a PEM single-cell electrolyzer, the IrO$_2$@TaB$_2$ can achieve 3.06 A cm$^{-2}$ current density at 2.0 V and 80 °C (membrane: Nafion$^{TM}$ N115), with a low iridium-loading of 0.15 mg cm$^{-2}$. The performance of our PEMWE has reached that of US DOE 2023 target (1.9 V@2.5 A cm$^{-2}$, platinum group metal (PGM) loading of 1 mg cm$^{-2}$), under a much lower PGM loading (0.15 mg$_{Ir}$ cm$^{-2}$ and 0.27 mg$_{Pt}$ cm$^{-2}$).

## Results

### Fabrication and characterizations of nano-metal diborides

We achieve the structural transformation of metal disulphide into metal diboride (MS$_2$ + B → MB$_2$ + S$_x$) under a KCl–CsCl molten salt condition (Fig. 1a and Supplementary Table 1). The gaseous sulfur-containing products may consist of S$_2$, S$_4$, S$_8$, and B$_2$S$_3$ at the reaction temperatures (1000–1100 °C), while S$_2$ is considered to be the most abundant species[31]. We calculate the reaction enthalpy and entropic term to determine the Gibbs free enthalpy of the chemical reaction ($\Delta G = \Delta H - T\Delta S$). As exhibited in Fig. 1b and Supplementary Table 2, the entropic term T$\Delta S$ of the gaseous sulfur-containing products is the dominant part of the total Gibbs energy. After reaching a certain temperature, the entropy gain would drive the reaction from a thermodynamically unfeasible process to a thermodynamically spontaneous process. According to the entropy-driven route, we can synthesize nine monometallic diborides, including MB$_2$ (M = Ti, Zr, Hf, V, Nb, Ta, Mo, W, and Re). The powder X-ray diffraction (XRD) patterns in Fig. 1c confirm that the majority of as-synthesized samples are pure transition metal diboride phases. The morphologies of these borides are revealed by scanning electron microscopy (SEM, Supplementary Fig. 2). TaB$_2$ and MoB$_2$ possess a typical nanosheet structure, WB$_2$

shows a nanowire morphology, and other MB$_2$ samples (M = Ti, Zr, Hf, V, Nb, and Re) are composed of nanoparticles with sizes ranging from 50 to 200 nm. Most of these MB$_2$ samples possess large Brunauer–Emmett–Teller (BET) surface areas within the range of 40–60 m$^2$ g$^{-1}$ (Fig. 1d and Supplementary Table 3), which are obviously higher than those of borides obtained by conventional high-temperature ceramic method[32–36].

We have compared the acid corrosion resistance and BET surface area of nine metal diborides synthesized above. Compared with TaB$_2$, the TiB$_2$, ZrB$_2$, HfB$_2$, VB$_2$, CrB$_2$, MoB$_2$, and WB$_2$ show more severe metal dissolution in acid (Supplementary Table 3). After the exposure to 0.5 M H$_2$SO$_4$ over a week (Supplementary Fig. 3), almost no Ta species are leached in the acidic solution, and TaB$_2$ retains its pristine crystal structure well. These results imply the outstanding chemical stability of TaB$_2$ in acid. In addition, TaB$_2$ presents much higher BET surface area than HfB$_2$, VB$_2$, NbB$_2$, and ReB$_2$ (Supplementary Fig. 3). From the crystal structure perspective, the TaB$_2$ comprises 3D metallic Ta–Ta framework and 2D graphene-like boron layers (also known as borophene subunits), suggesting a fast electron transport property[37]. Experiments present TaB$_2$ has a high conductivity of 25.2 S cm$^{-1}$ (Supplementary Fig. 4), which is significantly larger than those of previously-reported supports of IrO$_2$[38–41], such as TiN (3.9 S cm$^{-1}$), TaC (0.65 S cm$^{-1}$) and TiO$_2$ (1.8 × 10$^{-4}$ S cm$^{-1}$). Given its multi-advantages including large BET surface areas, great acid corrosion resistance, and high conductivity, the nano-TaB$_2$ is specially selected to explore the potential as support of IrO$_2$ catalysts. Moreover, the crust abundance of Ta is five orders of magnitude higher than that of Ir (Supplementary Table 4), and Ta costs around 2‰ of the Ir price ($367.5 for Ta vs. $164,662.0 for Ir per kilogram in 2023). Hence, the introduction of TaB$_2$ supporting material is expected to reduce the cost of the anode catalyst layer and improve the feasibility of PEMWE.

We further characterized the TaB$_2$ sample with SEM and transmission electron microscopy (TEM). The SEM (Supplementary Fig. 2f) and TEM images (Fig. 1e) present hat TaB$_2$ nanosheets possess an edge length of 200–500 nm and a thickness of 5–20 nm. High-resolution TEM image (HRTEM) of TaB$_2$ nanosheet (Fig. 1f) reveals its good crystallinity. In the HRTEM image and the corresponding selected-area electron diffraction (SAED) pattern (Fig. 1f, inset), interplanar spacing of 0.268 nm could be observed for three sets of lattice fringes, assigning to {001} planes of TaB$_2$ phase. The angle of 60° between the three planes is consistent with the theoretical value. Moreover, N$_2$ adsorption-desorption isotherms of TaB$_2$ display typical type-II curves with H3 hysteresis loop (Fig. 1d), suggesting the presence of inter-laminar pores among adjacent nanosheets. The BET surface area of TaB$_2$ is calculated to be 54.6 m$^2$ g$^{-1}$. These results demonstrate the successful synthesis of high-quality TaB$_2$ nanosheet with large surface areas.

### Loading IrO$_2$ nanocatalysts on boride supports

The loading of IrO$_2$ nanocatalysts on TaB$_2$ supports (IrO$_2$@TaB$_2$) is achieved through calcinating a mixture of K$_2$IrCl$_6$ and TaB$_2$ in molten NaNO$_3$ (i.e., 350 °C). The loading amounts of IrO$_2$ can be controlled by changing the feeding of the iridium source. When TaB$_2$ is not employed in the synthesis, pure IrO$_2$ nanoparticles with a size of 1.4–2.0 nm are formed (Supplementary Fig. 5 and 6). A series of IrO$_2$@TaB$_2$ samples with Ir contents of 4 wt%, 10 wt%, 16 wt%, 25 wt%, and 40 wt% are synthesized, and their iridium contents are quantified by inductively coupled plasma atomic emission spectroscopy (ICP-OES). The XRD patterns of these IrO$_2$@TaB$_2$ samples are shown in Fig. 2a, with those of TaB$_2$ and unsupported IrO$_2$ as references. The diffraction peaks of TaB$_2$ supports dominate the XRD patterns of IrO$_2$@TaB$_2$. In addition, a broadened peak at 30-40° can be attributed to IrO$_2$, suggesting the very small size of IrO$_2$. With the increase in IrO$_2$ contents on TaB$_2$ supports, the diffraction intensities of IrO$_2$ increase gradually.

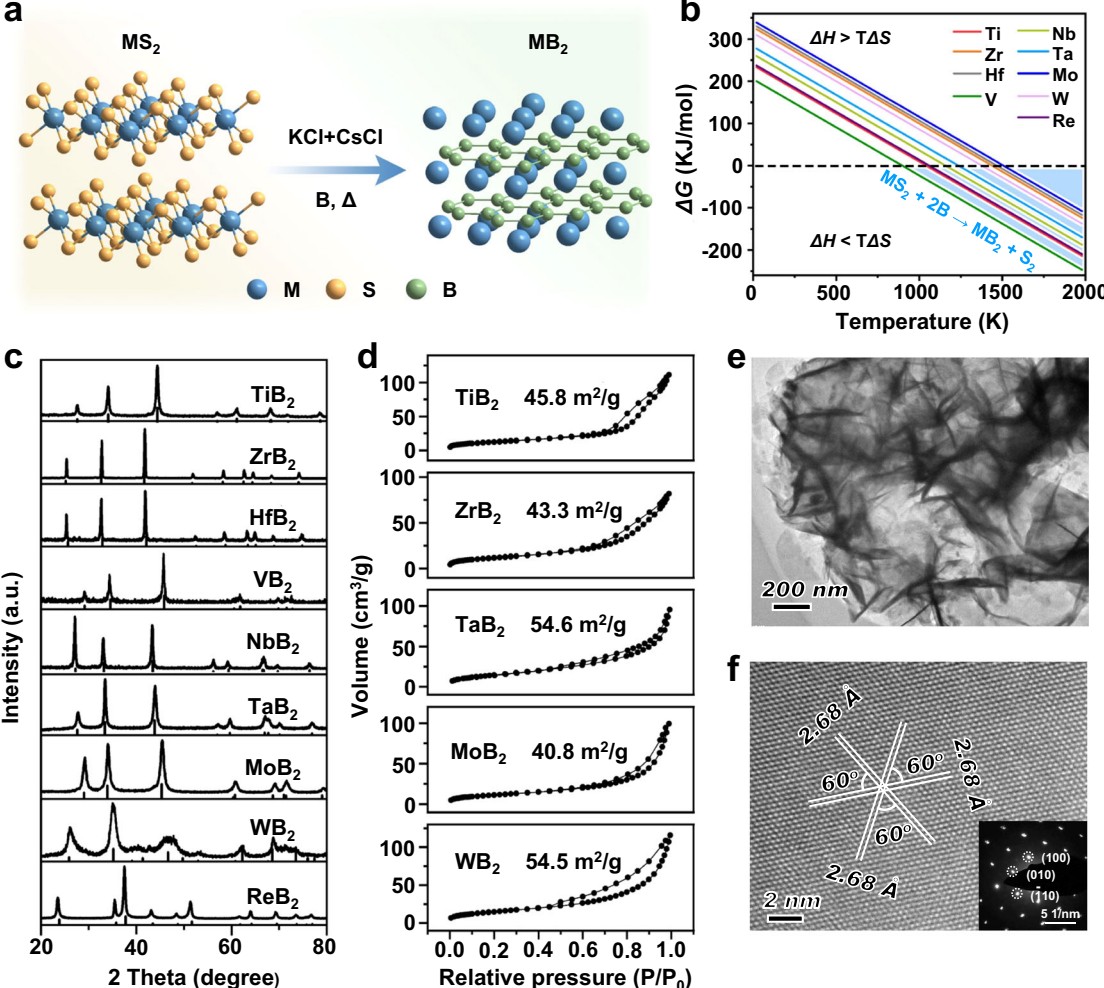

**Fig. 1 | Fabrication and structural characterizations of nano-metal diborides.**
**a** A schematic presenting the synthesis and crystal structure of TaB$_2$.
**b** Temperature-dependent Gibbs free energy of the synthesis process of metal diborides. **c** XRD patterns of as-synthesized diborides samples. For comparison, the
Joint Committee on Powder Diffraction Standard (JCPDS) cards of these metal diborides are included. **d** N$_2$ adsorption-desorption isotherms of diborides samples. **e** TEM image and (**f**) HRTEM image of TaB$_2$. The inset of Fig. 1f shows SAED pattern of TaB$_2$.

These IrO$_2$@TaB$_2$ samples are further characterized by X-ray photoelectron spectroscopy (XPS) and X-ray absorption spectroscopy (XAS). Figure 2b shows Ta4$f$ XPS spectra of IrO$_2$@TaB$_2$ samples and the reference TaB$_2$. For TaB$_2$, the surface comprises both metallic Ta elements and oxidized Ta elements, indicating slight surface oxidation. Unlike that of TaB$_2$, the surface layers of IrO$_2$@TaB$_2$ samples are fully dominated by oxidized Ta elements, suggesting further surface oxidation of Ta species during the IrO$_2$ loading process. With increasing IrO$_2$ content in the samples, the signal intensities of Ta4$f$ XPS decrease gradually, implying the decrease of surface Ta concentration. Similarly, the surface B concentration also decreases with increasing IrO$_2$ content and is almost undetectable after the 16% Ir content (Fig. 2c). Based on the Ta, B and Ir XPS signal intensities (Fig. 2b–d), we conclude that the Ta and B species dominate the surfaces of IrO$_2$@TaB$_2$ samples with the Ir loading content ≤10%; when the Ir loading content >10%, the Ta and B signal intensities are very weak and the Ir species dominate the surfaces, indicating a dense coating of IrO$_2$ formed at the support surface. In addition, the Ir4$f$ XPS peaks of IrO$_2$@TaB$_2$ samples (Fig. 2d) present an obvious shift to larger binding energy relative to those of IrO$_2$, and the deviation increases with the decrease in IrO$_2$ contents on TaB$_2$ supports. This indicates a slightly lower Ir oxidation in IrO$_2$@TaB$_2$ due to the electronic interaction between IrO$_2$ and support. The

electronic interaction and Ir oxidation state can be flexibly regulated by the Ir loading content.

The charge redistribution of IrO$_2$ is also supported by X-ray absorption near-edge structure (XANES) spectra. We note that we chose 16 wt% IrO$_2$@TaB$_2$ as a representative sample, which exhibits an optimized iridium mass activity toward OER in a standard three-electrode cell. Unless otherwise specified, subsequent IrO$_2$@TaB$_2$ refers to the sample with 16 wt% Ir loading. We also note that we chose Ir L$_1$-edge for the XANES measurements, considering that the Ir L$_3$-edge overlaps with the Ta L$_2$ and Ta L$_1$-edges. Figure 2e shows the Ir L$_1$-edge XANES spectra of IrO$_2$@TaB$_2$ and IrO$_2$. Compared to that of IrO$_2$, the white-line energy position of IrO$_2$@TaB$_2$ has a slightly weaker peak intensity, indicating an increased occupation in 5d electron orbital and a relatively lower Ir oxidation state in IrO$_2$@TaB$_2$. Moreover, extended X-ray absorption fine structure (EXAFS) exhibits similar structural information of IrO$_2$@TaB$_2$ and IrO$_2$, and the corresponding fitting results (Fig. 2f and Supplementary Table 5) show that both the Ir-Ir and Ir-O bond distances in IrO$_2$@TaB$_2$ are close to those in IrO$_2$. According to the above XPS and XAS results, we can reasonably conclude that: (i) the lattice of IrO$_2$ has almost unchanged after loading on TaB$_2$; (ii) however, the interfacial electronic interaction between IrO$_2$ and support leads to the charge redistribution of IrO$_2$.

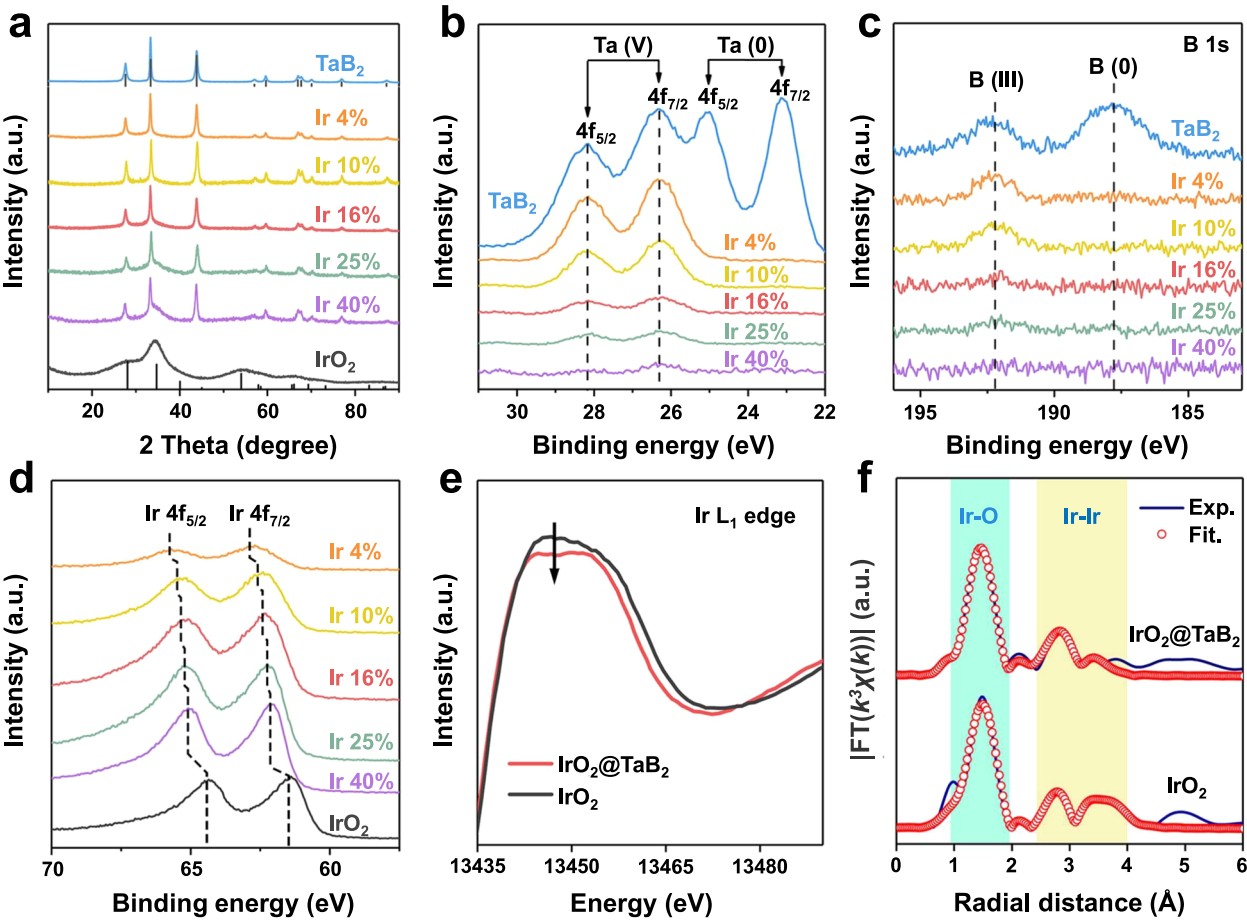

**Fig. 2 | Structural characterizations of $IrO_2@TaB_2$. a** XRD patterns, (**b**) Ta4$f$ XPS spectra, (**c**) B1$s$ XPS spectra, and (**d**) Ir4$f$ XPS spectra of $IrO_2@TaB_2$ samples with different $IrO_2$ loading. **e** Ir $L_1$-edge XANES spectra of 16 wt% $IrO_2@TaB_2$ and $IrO_2$. **f** Fourier transforms of the EXAFS spectra of 16 wt% $IrO_2@TaB_2$ and $IrO_2$.

The aberration-corrected high-angle annular dark-field scanning TEM (HAADF-STEM) was applied to investigate $IrO_2@TaB_2$ sample. As shown in Fig. 3a, the $IrO_2$ nanoparticles are uniformly and tightly dispersed on the $TaB_2$ sheets. The main sizes of $IrO_2$ are about 1.5 nm (Supplementary Fig. 6). The energy dispersive X-ray spectrum (EDX) elemental mappings of $IrO_2@TaB_2$ sample in Fig. 3b and Supplementary Fig. 7 confirm that Ta and B elements show sheet morphology, while Ir and O elements are homogeneously distributed in supports. The side view of TEM image (Fig. 3c) reveals that an around 5 nm surface coating of $IrO_2$ nanocrystallites is formed on $TaB_2$ supports. The amorphous filler among $IrO_2$ nanocrystallites can be attributed to $TaO_x$ layer on $TaB_2$ surfaces. Both the side view and top view of TEM images (Fig. 3c, d) show that the $IrO_2$ nanocrystallites are randomly but spatially interconnected. The HRTEM image (Fig. 3e) presents lattice fringes of the (11$\bar{1}$) plane (-0.224 nm) and (020) plane (-0.225 nm) of $IrO_2$, respectively. The angle between the two planes is determined as 59.1°, equal to the theoretical value.

Based on the above electron microscopy, electron diffraction, and spectroscopic evidence, the structural illustration of $IrO_2@TaB_2$ sample is shown in Fig. 3f. First, the surface of $TaB_2$ support tends to form amorphous $TaO_x$ layer. Second, the $IrO_2$ nanocrystallites are uniformly and tightly dispersed on the supports to form 5 nm thick $IrO_2$ layers. Third, the $IrO_2$ nanocrystallites with intrinsic conductive properties are spatially interconnected, and the $IrO_2$ is electrically connected to the metallic $TaB_2$ support, ensuring the formation of conductive networks. Fourth, the interfacial electronic coupling in $TaO_x/IrO_2$ catalytic layer results in the valence state reduction of Ir atoms.

## Electrocatalytic performance in a three-electrode configuration

The electrocatalytic activities toward OER of $IrO_2@TaB_2$ were measured using a standard three-electrode configuration in 0.1 M $HClO_4$ electrolyte. As the $IrO_2$ content increases, the electrocatalytic activity for OER increased rapidly (Fig. 4a), and reaches its optimization when the Ir content is 16 wt%. However, a further increase in the Ir content of the $IrO_2@TaB_2$ results in a slight deterioration of the electrocatalytic activity. The required overpotential to generate a current density of 10 mA cm$^{-2}$ (normalized over the electrode geometric area) for 16 wt% $IrO_2@TaB_2$ is 288 mV, lower than that of unsupported $IrO_2$ (307 mV). The Tafel slope of 16 wt% $IrO_2@TaB_2$ is 42.6 mV dec$^{-1}$ (Supplementary Fig. 8), close to that of unsupported $IrO_2$ (45.1 mV dec$^{-1}$), indicating that the the $TaB_2$ support has no change to OER pathway of $IrO_2$.

Electrochemical double layer capacitance (DLC) measurements were carried out to obtain the electrochemically active surface area (ECSA) of $IrO_2@TaB_2$ by estimating the accumulated charge amount at the electrode surface. Generally, the ECSA values become larger with increasing Ir content in $IrO_2@TaB_2$ (Fig. 4b, left and Supplementary Table 6). In order to compare the intrinsic activities of different catalysts, we normalized the measured currents by the ECSAs (Fig. 4b, right). The 16 wt% $IrO_2@TaB_2$ presents the highest current density ($j_{ECSA}$) at 1.53 V versus reversible hydrogen electrode (RHE) among these $IrO_2@TaB_2$ samples, which is 5.2 times larger than that of $IrO_2$. We further compared the iridium mass activities of $IrO_2@TaB_2$ and $IrO_2$ by normalizing the measured currents over the mass of iridium ($j_{ir}$). Although 16% $IrO_2@TaB_2$ has 70 wt% less iridium relative to $IrO_2$, it displays a high iridium mass activity of 345 A g$^{-1}$ at 1.53 V versus RHE

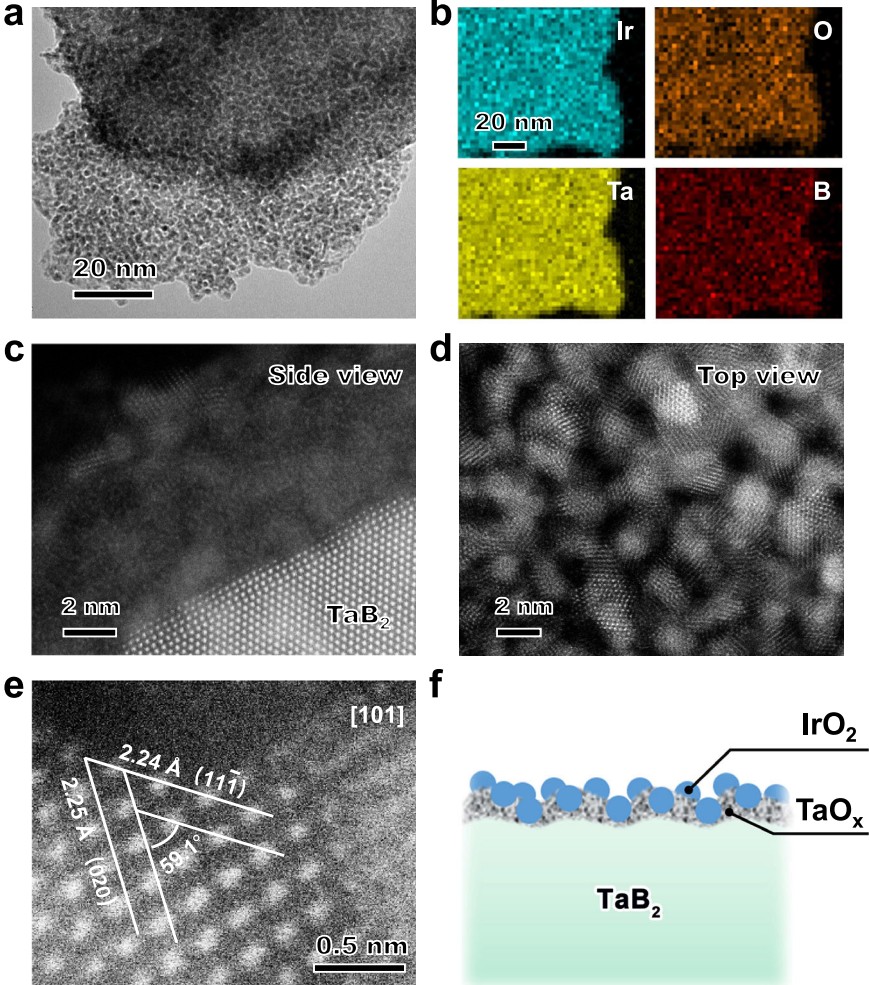

**Fig. 3 | Electron micrographs of IrO₂@TaB₂. a** The TEM image of IrO₂@TaB₂.
**b** The elemental mapping images of IrO₂@TaB₂. **c, d** The side view and top view of
aberration-corrected HAADF-STEM images of IrO₂@TaB₂. **e** High-resolution
HAADF-STEM image of IrO₂@TaB₂. **f** Schematic illustration of microstructure of
IrO₂@TaB₂.

(Fig. 4c and Supplementary Table 7), which is an order of magnitude higher than that of $IrO_2$. The iridium mass activity of 16 wt% $IrO_2@TaB_2$ is superior to some recently-reported iridate electrocatalysts (e.g., 3C-$SrIrO_3$, 6H-$SrIrO_3$, $Sr_2IrO_4$, $K_{0.25}IrO_2$)[42–47] and those representative supported catalysts (e.g., $IrO_2$-$TiO_2$, $IrO_2@Ir/TiN$, $IrO_2/Nb_{0.2}Ti_{0.8}O_2$)[40,48–52].

Besides high activity, $IrO_2@TaB_2$ displays great catalytic and structural stability for acidic OER. The galvanostatic measurements (Fig. 4d) demonstrate that $IrO_2@TaB_2$ remains steadily catalytic activity for more than 120 h of continuous operation at a current density of 10 mA cm$^{-2}$, while $IrO_2$ lost its catalytic activity after 40 h in the acidic electrolyte. Supplementary Fig. 9 exhibits the polarization curve for OER obtained from $IrO_2@TaB_2$ and $IrO_2$ before and after 5000 cycles. $IrO_2@TaB_2$ presents no measurable loss of catalytic activity after the continuous polarization measurements, while $IrO_2$ suffers from a severe deactivation. These results suggest that $TaB_2$ support greatly enhances the catalytic stability of $IrO_2$. Moreover, $IrO_2@TaB_2$ achieves a faradaic efficiency of nearly 100% during acidic OER (Supplementary Fig. 10), confirming that the observed current can be entirely attributed to the oxygen generation.

The leached amounts of cations during OER of $IrO_2@TaB_2$ and $IrO_2$ were quantitatively determined by ICP-OES (Fig. 4e). There are no detectable leached Ta and B species for $IrO_2@TaB_2$ in the electrolyte, and the Ir dissolutions increase during the first two hours and then reaches a stable state. The constant Ir concentration for $IrO_2@TaB_2$ in the electrolyte is 0.25 mg L$^{-1}$, which is much weaker than that for $IrO_2$ (0.4–0.7 mg L$^{-1}$), suggesting that the dispersion of $IrO_2$ on $TaB_2$ supports markedly improves the structural stability. The stability of $IrO_2@TaB_2$ is further evaluated by calculating the stability number (i.e., S-number proposed by Gieger et al.)[53], which is a good metric that relates the amount of evolved oxygen to the dissolved iridium. As shown in Fig. 4e (inset), the S-number of $IrO_2@TaB_2$ during the acidic OER is $5.2 \times 10^4$, which is higher than that of $IrO_2$ ($2.9 \times 10^4$). In addition, $IrO_2@TaB_2$ still maintains the initial morphology and structure after OER, as supported by HRTEM, XRD, and XPS results (Supplementary Fig. 11–13). These results overall confirm the excellent electrocatalytic and structural stability of $IrO_2@TaB_2$ toward acidic OER.

## Catalytic mechanism and origin of high activity

To investigate the catalytic mechanism, OER performance of $IrO_2@TaB_2$ and $IrO_2$ were examined in $HClO_4$ electrolytes with different pH. The catalytic activities of $IrO_2@TaB_2$ and $IrO_2$ show strong pH dependence in the pH range of 0–1.5 on the standard hydrogen electrode (SHE) scale (Fig. 5a). The potential–pH dependence value of $IrO_2@TaB_2$ is −57.3 mV dec$^{-1}$, which is near to that of $IrO_2$ (−58.0 mV dec$^{-1}$, Fig. 5b). Such a near −60 mV dec$^{-1}$ Nernstian potential shift

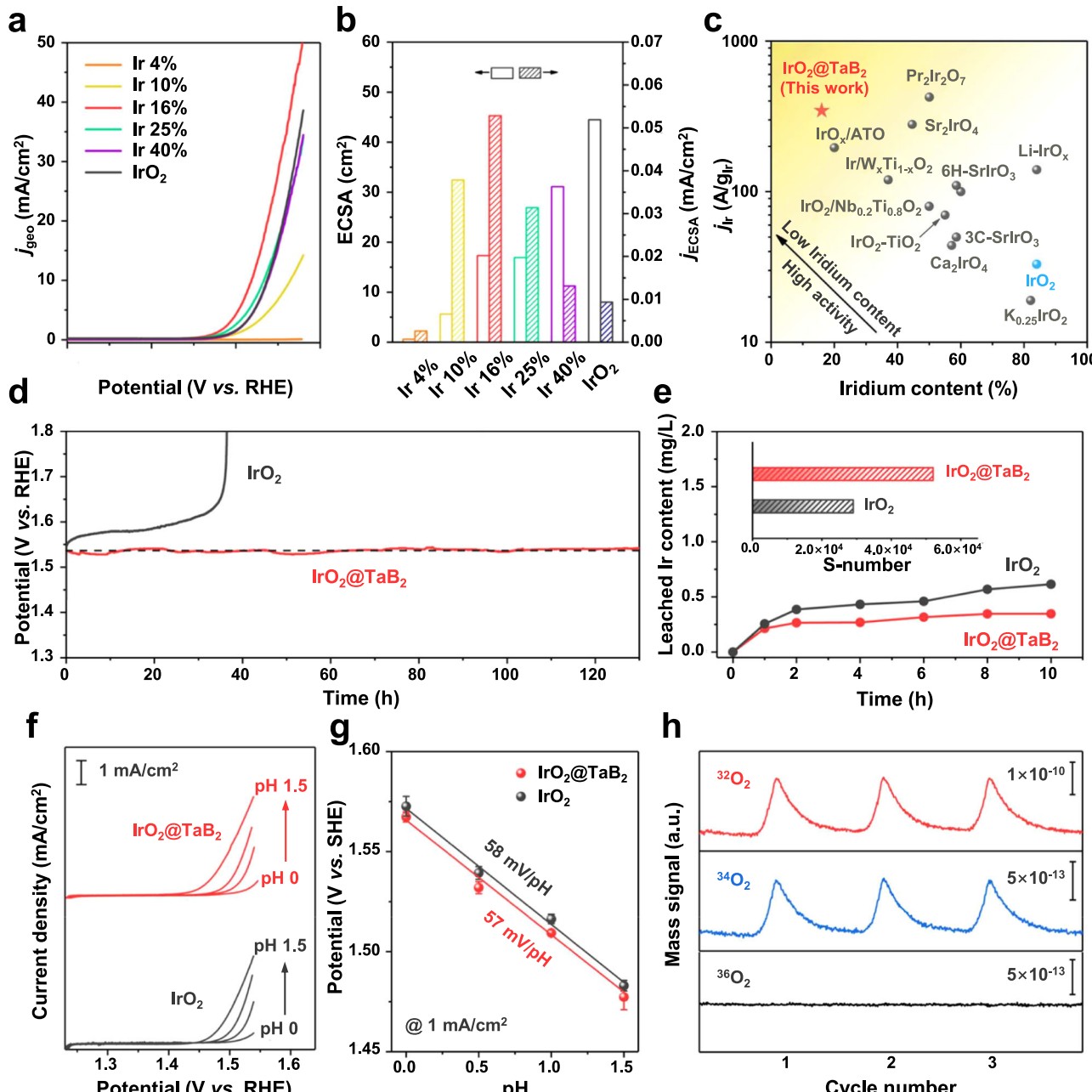

**Fig. 4 | OER performance and catalytic mechanism. a** The polarization curves toward OER in the presence of IrO$_2$@TaB$_2$ and IrO$_2$ in 0.1 M HClO$_4$. **b** Comparison of ECSA and current densities ($j_{ECSA}$) normalized by ECSAs for IrO$_2$@TaB$_2$ and IrO$_2$. **c** Comparison of iridium mass activities ($j_{Ir}$) of Ir-based electrocatalysts at 1.53 V versus RHE[40,42–52]. **d** Chronopotentiometric curves of IrO$_2$@TaB$_2$ and IrO$_2$ with a current density of 10 mA cm$^{-2}$. **e** Contents of leached iridium in the electrolyte in the presence of IrO$_2$@TaB$_2$ and IrO$_2$ during electrocatalysis. **f** The polarization curves for OER of IrO$_2$@TaB$_2$ and IrO$_2$ in HClO$_4$ electrolyte with different pH. **g** pH dependence of IrO$_2$@TaB$_2$ and IrO$_2$ on the OER potential on the SHE scale. **h** DEMS signals of O$_2$ products for $^{18}$O-labeled IrO$_2$@TaB$_2$ in 0.1 M HClO$_4$ in H$_2$$^{16}$O.

indicates a proton coupled electron transfer (CPET) process[54], and the OER at Ir sites strictly follows the conventional adsorbates evolution mechanism (AEM) pathway. In addition, we applied in situ differential electrochemical mass spectrometry (DEMS) to study the reaction mechanisms. The O source of evolved oxygen product can be identified through labeling of catalysts with $^{18}$O isotope. When the $^{18}$O-labeled IrO$_2$@TaB$_2$ electrocatalyst works in H$_2$$^{16}$O electrolyte, $^{34}$O$_2$ (or $^{16}$O$^{18}$O) to $^{32}$O$_2$ ratio in gaseous product is 0.43% (Fig. 5c). It should be noted that the natural stable abundance of $^{18}$O isotope is about 0.2% in water[55,56], indicating the minimum detected amount of $^{34}$O$_2$ in OER products is about 0.4%. We further carried out DEMS measurement for $^{18}$O-labeled IrO$_2$ catalyst under the same condition, the $^{34}$O$_2$/$^{32}$O$_2$

intensity ratio from the reaction products is also 0.43% (Supplementary Fig. 14). These results overall demonstrate that like that of well-known IrO$_2$ catalysts, the OER mechanism of IrO$_2$@TaB$_2$ catalyst strictly undergoes the adsorbate evolution mechanism (AEM), excluding the participation of lattice oxygen during the electrocatalysis.

After determining the catalytic mechanism, we sought to explain why IrO$_2$@TaB$_2$ possesses high activity for OER. Given that the crystal lattice and catalytic mechanism of IrO$_2$ have almost no change after loading on TaB$_2$, the enhanced performance of IrO$_2$@TaB$_2$ for OER is interpreted as follows. (i) The average particle sizes of IrO$_2$@TaB$_2$ and IrO$_2$ are estimated to be 1.5 nm and 1.8 nm (Supplementary Fig. 6).

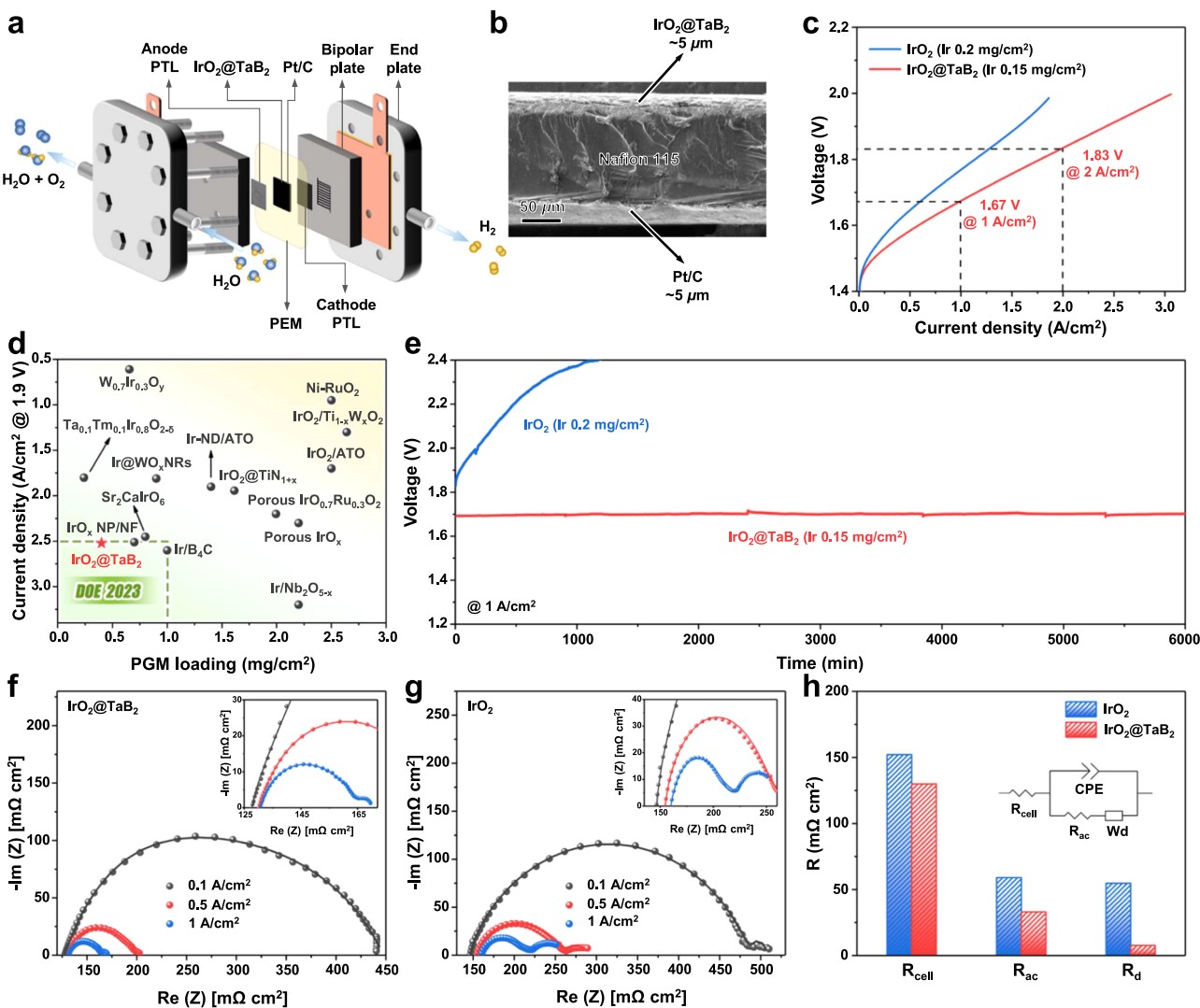

**Fig. 5 | Performance of PEMWE devices. a** Stack structure and key materials of a PEMWE. **b** Cross-section SEM image of CCM employing 40% $IrO_2@TaB_2$ anode layer and 40% Pt/C cathode layer. **c** Polarization curves of PEMWEs using $IrO_2@TaB_2$ and $IrO_2$ anodes at 80 °C with Nafion 115 membrane. **d** Comparison of current densities of PEM electrolyzers using different iridium-based catalysts at a cell potential of 1.9 V[16,18,60–69]. **e** Chronopotentiometry curve of PEMWEs using $IrO_2$ and $IrO_2@TaB_2$ anodes operated at 1 A cm$^{-2}$. **f, g** EIS curves of PEMWEs using $IrO_2@TaB_2$ and $IrO_2$ anodes. **h** The comparison of ohmic resistance, activation resistance, and diffusion resistance for PEMWEs using $IrO_2@TaB_2$ and $IrO_2$ anodes. The inset shows EEC model for EIS fitting.

Confinement of $IrO_2$ nanoparticles on the $TaB_2$ supports reduces $IrO_2$ size, which provides more active sites to improve OER activity. (ii) The interfacial electronic coupling in $TaO_x/IrO_2$ catalytic layer is responsible for the high intrinsic activity of $IrO_2@TaB_2$. A metal−semiconductor heterojunction is constructed between $TaO_x$ and $IrO_2$ on $TaB_2$ surface, resulting in the formation of a surface electric field and strong electronic interaction (Supplementary Fig. 15a). The electrons will flow from the conduction band of $TaO_x$ to $IrO_2$, driven by the work function differences, leading to electron-rich $IrO_2$. The antibonding states of $IrO_2$ are more fully occupied by electrons in the $IrO_2$-$TaO_x$ heterojunction, which lowers the surface oxygen adsorption of $IrO_2$ and consequently boosts the OER activity (Supplementary Fig. 15b, c). We note that the chemical and structural complexity of $IrO_2@TaB_2$ catalysts, including the uncertain surface structures of $IrO_2$ nanoparticles (without specific exposed facets), amorphous $TaO_x$ layer, and their heterointerface, makes it unrealistic to establish a clear structural model of the catalyst. This limits us from accurately modeling the electrochemical processes of OER and quantitatively describing catalytic activity by advanced calculation methods (e.g., grand-canonical DFT)[57].

## Performance of PEMWE devices

The performance of as-prepared $IrO_2@TaB_2$ as an anode catalyst was finally evaluated on a real PEM electrolyzer (Fig. 5a). Unlike the three-electrode configuration to mainly reflect the performance of the catalyst itself, the PEMWE requires excellent electrical conductivity of catalyst (greater than 0.1 S cm$^{-1}$) to deliver high current densities of several A cm$^{-2}$ [3,4]. The electrical conductivity of $IrO_2@TaB_2$ becomes higher as the $IrO_2$ content increases (Supplementary Fig. 16), and up to 0.17 S cm$^{-1}$ for the 40% $IrO_2@TaB_2$ sample, which is similar to the conductivity of $IrO_2$ itself (0.18 S cm$^{-1}$). The structure and composition characterizations of the 40%$IrO_2@TaB_2$ sample by ICP-OES, XRD, XPS (Fig. 2a–c), and TEM (Supplementary Fig. 17) confirm that there is no essential difference between 40%$IrO_2@TaB_2$ and 16%$IrO_2@TaB_2$ sample, except for higher Ir loading of the former. Hence, the 40% $IrO_2@TaB_2$ sample is employed as the optimal anode catalyst to integrate into PEMWE. The Nafion 115 membrane is used as the PEM, which is a perfluorosulfonic polymer with a thickness of 125 μm. The CCM with a 5 cm$^2$ working area (Supplementary Fig. 18) employing 40% $IrO_2@TaB_2$ anode layer and 40% Pt/C cathode layer is produced by a decal transfer method. SEM images in Fig. 5b and Supplementary

Fig. 19 exhibit the cross-section and top view morphology of the CCM. Both the catalyst particles of $IrO_2@TaB_2$ anode and Pt/C cathode are composed of uniformly distributed agglomerates on the membrane surface. The thickness of both anode and cathode catalyst layers on the membrane is roughly 5 μm. As analyzed by ICP-OES, the CCM contains a low Ir loading of 0.15 mg cm$^{-2}$ at the anode layer and a low Pt loading of 0.27 mg cm$^{-2}$ at the cathode layer (Supplementary Fig. 20), respectively. The CCM shows lower total loadings of PGMs (0.42 mg cm$^{-2}$) than the DOE 2023 target (1.0 mg cm$^{-2}$) and even the DOE 2025 target (0.5 mg cm$^{-2}$)[58].

Even such low noble metals used in our CCM, the polarization curve of PEMWE reveals a current density of 3.06 A cm$^{-2}$ with a cell potential of 2.0 V, operating at 80 °C and ambient pressure (Fig. 5c). It is surprising that the performance of our PEMWE has reached that of US DOE 2023 target (1.9 V@2.5 A cm$^{-2}$), under such low PGM loadings (0.15 mg$_{Ir}$ cm$^{-2}$ and 0.27 mg$_{Pt}$ cm$^{-2}$). We note that the traditional CCMs in commercial PEMWE achieve reasonable activity and stability using a high Ir loading of 2–4 mg$_{Ir}$/cm$^2$ to ensure sufficient in-plane conductivity and mechanical stability of catalyst layer[59]. In fact, the performance of PEMWE using $IrO_2@TaB_2$ as anode electrocatalyst exceeds the most recent reports of PEMWEs using novel anode electrocatalysts (e.g., $Sr_2CaIrO_6$, $Ta_{0.1}Tm_{0.1}Ir_{0.8}O_{2-δ}$, $Ir@WO_x$)[16,18,60–67], as shown in Fig. 5d and Supplementary Table 8. Even if several supported catalysts (e.g., $Ir@B_4C$, $Ir@Nb_2O_{5-x}$) have reported to reach the DOE 2023 target, they require several times higher iridium loadings[68,69].

We further compare the performance of PEMWEs using $IrO_2@TaB_2$ and commercial $IrO_2$ as anode electrocatalysts (Fig. 5c), under the similar Ir loading in CCM. While $IrO_2$ employing as anode layer, the PEMWE delivers a significantly lower current density of 1.9 A cm$^{-2}$ relative to $IrO_2@TaB_2$ catalyst (3.06 A cm$^{-2}$) at 2. 0 V cell potential. To evaluate catalyst stability, the $IrO_2$ and $IrO_2@TaB_2$ anode cells were tested at a constant current density of 1 A cm$^{-2}$. The PEMWE using $IrO_2$ anode undergoes a severe deactivation under the low Ir loading of 0.2 mg cm$^{-2}$, and the $IrO_2$ anode cell exhibits reasonable stability when the Ir loading increases to 2 mg cm$^{-2}$ (Fig. 5e and Supplementary Fig. 21–24). By comparison, the PEMWE using $IrO_2@TaB_2$ anode provides a steady operation for more than 120 h under a low Ir loading of 0.15 mg cm$^{-2}$ (Fig. 5e), confirming excellent catalytic stability. We can speculate that the $IrO_2@TaB_2$ with larger material volume can forms a thicker catalyst layer relative to bare $IrO_2$, so that the former possesses sufficient layer conductivity and stability at such a low Ir loading. The great activity and stability demonstrate the great potential of $IrO_2@TaB_2$ as a practical anode of real PEMWE for commercial application.

To deduce the origin of performance differences, we carry out chemical impedance spectroscopy (EIS) experiments on PEMWEs using $IrO_2@TaB_2$ and $IrO_2$ as anode electrocatalysts, respectively (Fig. 5f, g). We further fit the EIS Nyquist plots by equivalent electrical circuit (EEC) model (Fig. 5h, inset), which shows good accordance with the experimental results. The total losses in a PEMWE mainly compose of ohmic resistance, activation resistance, and diffusion resistance[70,71]. (i) The $R_{cell}$ denotes the cell ohmic resistance, which is related to ohmic losses of all components including membrane, catalyst layers, porous transport layers (PTL), bipolar plate, and their interfacial resistances. The $R_{Cell}$ is almost unaffected at different current densities for $IrO_2@TaB_2$ anode cell, which can be reflected by high frequency resistance (HFR, Supplementary Fig. 25). The $IrO_2@TaB_2$ anode cell shows a slight decrease in ohmic resistance relative to the $IrO_2$ anode cell. (ii) The $R_{ac}$ denotes the activation resistance to determine the reaction kinetics of anode and cathode electrocatalysts. $R_{ac}$ is sensitive to current density and is mainly contributed from the anode. The $R_{ac}$ of $IrO_2@TaB_2$ anode cell is significantly reduced compared with that of $IrO_2$ anode cell, due to the much better catalytic performance $IrO_2@TaB_2$ relative to $IrO_2$. (iii) The $R_d$ denotes the diffusion resistance, as reflected by the Warburg diffusion element (Wd). $R_d$ is negligible at a low current density and increases with the increase of current density. The $IrO_2@TaB_2$ anode cell presents much lower transport loss compared with the $IrO_2$ anode cell. Taken overall, it can be concluded that the $IrO_2@TaB_2$ anode cell exhibits simultaneous decreases in both ohmic, activation, and diffusion losses relative to the $IrO_2$ anode cell.

## Discussion

In conclusion, we have demonstrated an entropy-driven disulphide-to-diboride transition route strategy that can be operated to synthesize large-surface-area metal diborides as the promising support of $IrO_2$ nanocatalysts. As a demonstration, we have prepared $TaB_2$-supported $IrO_2$ nanocatalyst (i.e., $IrO_2@TaB_2$), which displays uniformly distributed $IrO_2$ nanoparticles (~1.5 nm), highly conductive networks, and great acidic corrosion resistance. Impressively, the as-prepared $IrO_2@TaB_2$ catalysts with an optimized $IrO_2$ content exhibit a 10 times higher iridium mass activity than $IrO_2$, while showing lower iridium leaching during acidic OER. When integrated into a PEMWE, the $IrO_2@TaB_2$ anode cell requires 2.0 V to attain 3.06 A cm$^{-2}$ (reaching the US DOE 2023 target), under a much lower PGM loadings of 0.15 mg$_{Ir}$ cm$^{-2}$ and 0.27 mg$_{Pt}$ cm$^{-2}$ than DOE 2023 target (PGM loading of 1 mg cm$^{-2}$). Our findings highlight the rational design of highly conductive, corrosion resistance, and large surface area materials as $IrO_2$ supports for acidic OER, and stimulate practical applications of low-iridium-loading anode catalyst layers in industrial PEMWE.

## Methods
### Chemicals and reagents
Absolute ethanol ($C_2H_6O$), isopropyl alcohol (($CH_3$)$_2$CHOH) and $NaNO_3$ were purchased from Sinopharm Chemical Reagent Co., Ltd. $TiS_2$, $ZrS_2$, $HfS_2$, $VS_2$, $NbS_2$, $TaS_2$ and $ReS_2$ were purchased from Nanjing MKNANO Tech. Co., Ltd. $MoS_2$ and $WS_2$ were purchased from Shanghai Aladdin Biochemical Technology Co., Ltd. $HClO_4$, was purchased from Tianjin Xinyuan Chemical Co., Ltd. Nafion® perfluorinated resin solution was purchased from Sigma-Aldrich. $K_2IrCl_6$ and $IrO_2$ were purchased from Shanghai Macklin Biochemical Co., Ltd. Pt/C (Pt 40 wt%) was purchased from Johnson Matthey Company. All the chemicals and reagents were used without further purification. Highly purified water (>18 MΩ cm resistivity) was obtained from a PALL PURELAB Plus system.

### Materials synthesis
The metal diborides were synthesized by borothermal reduction in a molten salt medium (KCl-CsCl) with an equal molar ratio. The metal disulfide, boron powder, and molten salt are ground thoroughly in a mortar for 0.5 h. Then the mixture was transferred into a corundum boat and heated at 900–1100 °C under Ar atmosphere in a tubular furnace, and the heating rate was 10 °C/ min (Supplementary Table 1). After thorough cooling, the mixture was repeatedly washed with water and ethanol to remove soluble molten salts and byproducts. The final prepared metal diborides were dried at 80 °C for 12 h.

For the Preparation of $IrO_2@TaB_2$, $K_2IrCl_6$ was dissolved in isopropyl alcohol, then $NaNO_3$ and prepared $TaB_2$ support were added to the solution. The suspension was ultrasonically dispersed for 30 min and then dried at 90 °C for 5 h. Then the mixed powder was ground and heated in a muffle furnace at 5 °C/min to 350 °C for 1 h. After naturally cooling down, the product was washed fully with ethanol and water to remove residual salts, and the black catalyst powder was dried in an oven at 90 °C for 12 h to get $IrO_2@TaB_2$.

For the preparation of unsupported $IrO_2$, the experimental process is the same as that for the preparation of $IrO_2@TaB_2$, except that $TaB_2$ support was not added.

### Characterizations
XRD patterns were conducted by a Rigaku D/Max 2550 X-ray diffractometer with Cu Kα radiation (λ = 1.5418 Å). SEM images were

obtained with field emission scanning electron microscopy (FESEM, JEOL 7800 F) at an accelerating voltage of 5 kV. EDX analysis was obtained with an EDX system attached to JEOL JSM7800F SEM. Low-magnification transmission electron microscope (TEM) images were obtained with a Philips-FEI Tecnai G2S-Twin microscope equipped with a field emission gun operating at 200 kV. High-resolution TEM images were recorded on a JEM-2100F electron microscope (JEOL, Japan). XPS measurements were carried out using a Thermo Fisher Scientific ESCALAB 250Xi with photoelectron spectroscopy system using a monochromatic Al Ka (1486.6 eV) X-ray source. ICP-OES was performed on a PerkinElmer Optima 3300DV ICP spectrometer. XAS (Ir $L_1$-edge) were collected at BL14W beamline in Shanghai Synchrotron Radiation Facility (SSRF). The storage rings of SSRF were operated at 3.5 GeV with a stable current of 200 mA. Using Si (111) double-crystal monochromator, the data collection was carried out in fluorescence mode using Lytle detector. All spectra were collected in ambient conditions. Data reduction, data analysis, and EXAFS fitting were performed with the Athena and Artemis software packages. The energy calibration of the sample was conducted through a standard $IrO_2$, which as a reference was simultaneously measured. For EXAFS modeling, EXAFS of the $IrO_2$ is fitted and the obtained amplitude reduction factor $S_0^2$ value (0.900) was set in the EXAFS analysis to determine the coordination numbers (CNs) in the Ir-O/Ir/Ta scattering path in the sample.

## Electrochemical measurements in a three-electrode configuration

The performances of catalysts were studied in a standard three-electrode system by a CH Instrument (Model 660E) and the electrolyte was 0.1 M $HClO_4$ solution bubbled with $O_2$ gas. Linear sweep voltammetry (LSV) measurements were performed with the scan rate of 1 mV s$^{-1}$ and compensated by 85% iR-drop. The chronopotentiometric curve was measured without iR-drop compensation. The counter electrode was a Pt wire and the reference electrode was a saturated calomel electrode (SCE). The potential was normalized with respect to the reversible hydrogen electrode (RHE), according to Eq. (1):

$$E_{vs.RHE} = E_{vs.SCE} + 0.249 eV + 0.059 pH \qquad (1)$$

where 0.249 eV vs. the SCE electrode is the potential of zero net current. The SCE reference electrode was calibrated before testing according to the method proposed by Boettcher and co-workers[72].

The working electrode was a glassy carbon electrode (GCE) loaded with catalysts. 4 mg catalyst powder was ultrasonically dispersed in 400 μL isopropyl alcohol containing 10 μL Nafion solution (5 wt.%). Then 2 μL catalyst ink was dropped on a GCE (Active area = 0.071 cm$^2$) and dried in air. The catalyst loading on the working electrode was 0.28 mg cm$^{-2}$. To avoid the passivation of GCE, a piece of carbon paper (Active area = 0.09 cm$^2$) loaded with catalysts was used as the working electrode for long-term stability test, and the catalyst loading was 0.3 mg cm$^{-2}$. In order to accurately evaluate the ion leaching and structural evolution behavior of the catalyst during the OER process, the catalyst ink was dropped onto the surface of a Ti plate (Active area = 1 cm$^2$) with a catalyst loading of 10 mg cm$^{-2}$. OER was catalyzed for 10 h at 10 mA cm$_{geo}^{-2}$, and a portion of the electrolyte was taken out every 2 h for ICP-OES. After the catalysis, the catalyst sample was obtained by eluting the electrode with ethanol.

To calculate $j_{geo}$ of catalysts, the measured current was normalized by the geometric area of GCE according to the Eq. (2):

$$j_{geo} = \frac{i}{s} (mA \, cm_{geo}^{-2}) \qquad (2)$$

where $i$ is the measured current, and $s$ is the geometric area of GCE.

To calculate $j_{Ir}$ of catalysts, the measured current was normalized by the mass of iridium supported on GCE according to the Eq. (3):

$$j_{Ir} = \frac{i}{m \times Ir(wt.\%)} (A \, gIr^{-1}) \qquad (3)$$

where m is the loading mass of catalysts on GCE, and Ir (wt.%) is the mass fraction of iridium in catalysts.

To calculate ECSA of catalysts, we performed CV tests in the non-faradaic current region between 0.83 V and 0.93 V vs. RHE at different scan rates (10, 25, 50, 75, and 100 mV s$^{-1}$). A linear trend could be obtained via plotting the currents difference (Δi) between the anodic and cathodic sweeps ($i_{anodic} - i_{cathodic}$) at 0.88 V against the scan rate. The slope of the fitting line was equal to twice the geometric double layer capacitance ($C_{dl}$). The ECSA of the catalyst was estimated according to the Eq. (4):

$$ECSA = \frac{C_{dl}}{C_s} (cm^2) \qquad (4)$$

where $C_s$ represents the specific capacitance of the catalyst (0.06 mF cm$^{-2}$).

To calculate $j_{ECSA}$ of catalysts, we normalized the meatured current by the ECSA from Eq. (5):

$$j_{ECSA} = \frac{i}{ECSA} (mA \, cm^{-2}) \qquad (5)$$

For estimation of Faradaic efficiency, a 2 h-long measurement was performed to investigate the Faradaic efficiency of catalysis with the galvanostatic experiment of 10 mA cm$^{-2}$. The oxygen gas released at every ten minutes was measured by the drainage collecting method. The mole of oxygen generated was calculated by the ideal gas law while the theoretical evolution of oxygen was calculated based on Faraday's law, assuming that all the electron transfer at the anode was from OER. The ratio of the actual molar amount of oxygen released to the theoretical molar amount of oxygen in the OER process was identified as the Faraday efficiency of $IrO_2$@TaB$_2$ electrocatalyst.

In situ DEMS test was performed in a three-electrode cell with 0.1 M $HClO_4$ solution as electrolyte. (i) The pristine $IrO_2$@TaB$_2$ was labeled with $^{18}$O isotope in 0.1 M $HClO_4$ solution using H$_2$$^{18}$O as solvent at 1.6 V for 10 min. (ii) The labeled electrode was washed with H$_2$$^{16}$O to remove the residual H$_2$$^{18}$O that was physically attached to the catalyst layer. Subsequently, CV tests were carried out at the rate of 50 mV s$^{-1}$ in the potential range of 0.6 V and 1.2 V vs. RHE for removing the adsorbed $^{18}$O species on the surface of the catalyst. (iii) Three LSV cycles were applied for the labeled electrode in 0.1 M $HClO_4$ solution using H$_2$$^{16}$O as a solvent in the potential range of 1.2–1.7 V vs. RHE. The gaseous products including $^{36}$O$_2$, $^{34}$O$_2$, and $^{32}$O$_2$ were monitored by the mass spectrometer. The fraction of $^{36}$O$_2$, $^{34}$O$_2$, and $^{32}$O$_2$ can be estimated from the integral areas of corresponding mass signals of these gaseous products.

## Electrochemical measurement of PEMWE

Before the construction of catalysis coated membrane (CCM), the N115 membrane was successively treated with 3 wt% $H_2O_2$, deionized water, and 0.5 M $H_2SO_4$ at 80 °C for 1 h. Then the treated N115 membrane was rinsed with deionized water. Commercial Pt/C (40 wt%) was used as a cathode electrocatalyst and commercial $IrO_2$ or $IrO_2$@TaB$_2$ (Ir 40 wt%) was used as an anode electrocatalyst. In order to prepare the catalyst ink, the catalyst was dispersed into a mixed solution of isopropyl alcohol and distilled water (1:1, w/w). Subsequently, Nafion with an ionomer mass fraction of 10 wt% at the anode or 35 wt% at the cathode was added into the solution. The suspension was ultrasonically treated in an ice water bath for 1 h to obtain the

catalyst ink. The anode catalyst ink and cathode catalyst ink were sprayed on polytetrafluoroethylene (PTFE) film respectively. Then the PTFE films supported with catalysts and a N115 membrane were hot pressed under 10 Mpa at 135 °C for 10 min. After cooling, the PTFE films were stripped to obtain CCM. The loading of the cathode was 0.27 $mg_{pt}$ $cm^{-2}$ and the loading of the anode was 0.15 $mg_{Ir}$ $cm^{-2}$ for $IrO_2@TaB_2$ or 0.2 $mg_{Ir}$ $cm^{-2}$ for $IrO_2$. The actual catalyst loading was determined by ICP-OES test. In order to construct a PEM electrolyzer, a well-defined pore Ti plate (provided by Hefei conservation of momentum green energy Co., Ltd) coated with Pt was used as the porous transport layer (PTL) of the anode, and a piece of carbon paper was used as the PTL of the cathode. The active area was 5 $cm^2$. The PEM electrolyzer was operated at 80 °C and the reactant was deionized water, which was circulated through a peristaltic pump. The polarization curve of the PEM electrolyzer was collected at the cell voltage of 1.4–2.0 V, and the stability was tested by chronopotentiometry at 1 A $cm^{-2}$.

## Computation details

We performed all DFT calculations using the Vienna ab initio simulation package (VASP 5.4.4)[73,74]. The Generalized gradient approximation (GGA) with the Perdew–Burke–Ernzerhof (PBE) exchange correlation functional was employed with 500 eV cut-off energy[75]. All crystal structures were obtained from the ICSD (Inorganic Crystal Structure Database) database. During the structural optimization, we used k-point separation length of 0.04 2π $A^{-1}$, while the k-point separation length was promoted to 0.03 2π $A^{-1}$ for DOS calculation. For all calculations, we apply an energy convergence criterion of $10^{-5}$ eV with the force convergence criterion of 0.02 eV $A^{-1}$.

For the Gibbs free energy change of the reaction, we use the Eq. (6) to calculate:

$$\Delta G = \Delta H - T\Delta S = \Delta E + \Delta E_{zpe} - T\Delta S \tag{6}$$

Where $\Delta E$ is the energy difference between product and reactant, $\Delta E_{zpe}$ is the zero-point energy difference between product and reactant, T is the absolute temperature, and $\Delta S$ is the entropy difference between product and reactant. For the reaction:

$$MS_2(s) + 2B(s) = MB_2(s) + S_2(g) \tag{7}$$

Since only $S_2$ is a gas, we only need to consider the zero-point energy and entropy of $S_2$. The former can be calculated as 4.47 kJ $mol^{-1}$, and the latter can be obtained as 228.2 J $mol^{-1}$ $K^{-1}$ from the database.

## Data availability

The data that support the findings of this study are available within the article and its Supplementary Information. All other relevant data supporting the findings of this study are available from the corresponding authors upon request.

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

## Acknowledgements
X.Z. and H.C. acknowledge funding from the National Key R&D Program of China (no. 2021YFB4000200), the National Natural Science Foundation of China (NSFC) (no. 22179046 and 22279040), the State Grid Headquarter Science and Technology project (5419-202158490A-0-5-ZN), the Jilin Province Science and Technology Development Plan (no. 20220402006GH and 20210101403JC), and Interdisciplinary Integration and Innovation Project of JLU (JLUXKJC2021ZZ18).

## Author contributions
X.Z. and H.C. directed this research. Y.W. conducted most of the experiments. M.Z. and L.S. performed the theoretical calculations. Z.K. contributed to the performance measurement of PEMWE devices. Y.S., B.T., and Y.Z. contributed to data analysis. X.Z. and H.C. wrote the paper. X.Z. and H.C. supervised the project. All authors discussed and reviewed the final manuscript.

## Competing interests
The authors declare no competing interests.
