## [Peer Review File · Nature Communications]

Nano-metal diborides-supported anode catalyst with strongly coupled TaOx/IrO₂ catalytic layer for low-iridium-loading proton exchange membrane electrolyzerREVIEWER COMMENTS

Reviewer #1 (Remarks to the Author):

The work deals with an investigation of tantalum diboride as effective support for iridium dioxide nanoparticles and its application as electrocatalyst for the oxygen evolution reaction in acidic medium. Noteworthy results comprise a high current density of 3 A cm⁻² at cell voltage of 2 V with a very low loading of iridium at the anode (0.15 mg cm⁻²). This result is of high significance, since it is positioned at the highest level in the field. The work presents an in-depth characterization of the catalyst structure, chemical composition and morphology, accompanied by small scale behavior (three electrode configuration and single cell 5 cm²) of catalytic activity, with durability assessment and stability. There are clear indications of the origin of activity, mechanistic insights and novel findings by DFT on the role of tantalum oxide phase in contributing to the activity of iridium nanoparticles.

Major concern:

- The main motivation to introduce a supporting material is to reduce the amount of the most critical active material, like iridium in this case, increasing the activity per mass of substance. One of the requirements of a proper support is to be abundant, inexpensive and resistant. Within the advantages claimed for boride ceramics, only physical characteristics are mentioned. In my opinion, the authors must comment on the scarcity and expensiveness of tantalum and tantalum borides compared to PGM in order to really assess the suitability of this approach to reduce cost and improve feasibility of proton exchange membrane water electrolyzers. How does the cost of tantalum diboride compare to iridium dioxide? What about other potential diborides (Ti, Zr, etc.)? What about abundance and economic importance?

Minor concerns:

- Calculate the ECSA by normalizing by the mass of active phase (m₂ gIr-1).
- What are the units of the values included in Supplementary Table 1?
- What conditions (more in particular potential window, scan rate, etc.) were used for the 5000 cycles (Supplementary Figure 9)?
- Did you test lower content than 0.27 mg cm⁻² of Pt at the cathode of the MEAs?
- In Figure 6e, the IrO₂-based MEA presents very low stability. Taking into account that iridium catalysts exhibit good durability of up to thousands of hours at even larger current density (doi: 10.1016/j.apenergy.2020.115809), what is the reason for low stability?
- Clearly indicate that the results depicted in figures 4c and 6d correspond to references of the supplementary information.
- Include a reference for the DOE target in PEMWE in 2023.
- Please, carefully revise the manuscript for typo errors. Just to name some of them: captions of figure 1 are wrong; line 108 'angle of 60°C'; line 146 'unsuppoted'; lines 160 and 162 'suraces'; lines 524 and 525 'menberane'; etc. I have not indicated all the typos I found during reading, for practical reasons, so please revise it thoroughly.

Reviewer #2 (Remarks to the Author):

The paper reports an interesting advance on the use of low Ir loading OER catalysts for PEM electrolysis. The subject is worthy of investigation as the high Ir loading is a major limitation to the mass deployment PEM electrolyzers. To achieve the target to cut the Ir loading down to 0.15 mg cm⁻² the authors propose the use of a heterojunction of IrO₂ with a TaB₂ support (Ta₂O₅ at the interface between the two component). The paper is well written and the conclusions are adequately supported by the deep characterization. I only have the following minor remarks:

- 1) What is the origin of the different stability of IrO₂ and IrO₂@TaB₂? Stability data are reported for 5

days. In this range the stability is excellent, but the stability the bare IrO₂ is extremely bad. Is this realistic? What is the origin of such dramatic difference?. I wonder if the total amount of material present in the electrode plays a role here. The volume of the catalyst that contains the TaB₂ is at least twice the volume of the bare IrO₂. I invite the author to comment on this in the manuscript.

2) The English is generally good, but careful proofreading is needed to check for typos here and there.

Reviewer #4 (Remarks to the Author):

The overall well-written manuscript by Chen, Zou and co-workers presents impressive performances for the oxygen evolution reaction via a carefully optimized IrO₂ catalyst supported on TaOx/TaB₂. The comparison with the literature is convincing and one can expect the developed catalyst to become a new benchmark. The work on the stability and characterization of the catalyst is particularly valuable.

The organization of the manuscript should be improved, as the choice of TaB₂ (as opposed to other borides) is justified after an already quite extensive presentation of this material.

Similarly, it is quite disappointing that the authors characterize and discuss in most detail the material with 16%IrO₂@TaB₂, but it is the 40%IrO₂@TaB₂ that is used for the actual device.

Given that the authors state themselves that the "TaB₂ support has no change to OER pathway of IrO₂", the theoretical "justification" of the activity of the hybrid system is weak/far fetched: (a) experimentally it is mostly the stability that is enhanced and not the intrinsic activity and (b) the argument of "electron-rich IrO₂" is not very plausible under oxidative conditions: In contrast to state-of-the-art modelling, the authors did not consider grand-canonical DFT, in other words, the experimentally applied electrochemical potential is disregarded in the computations. Under oxidative conditions, one can reasonably expect the surface to be (partially) positively charged to reach the relevant potentials (see, for instance, 10.1039/D0CP00281J). In short, the computational aspect of the manuscript in its present form does, alas, not provide significant added value to the work of the authors.

Detail:

The cited literature should include doi/10.1021/acscatal.0c04613.

Reviewer's Comments and Our Responses:

Reviewer #1 (Remarks to the Author):

The work deals with an investigation of tantalum diboride as effective support for iridium dioxide nanoparticles and its application as electrocatalyst for the oxygen evolution reaction in acidic medium. Noteworthy results comprise a high current density of 3 A cm^{-2} at cell voltage of 2 V with a very low loading of iridium at the anode (0.15 mg cm^{-2}). This result is of high significance, since it is positioned at the highest level in the field. The work presents an in-depth characterization of the catalyst structure, chemical composition and morphology, accompanied by small scale behavior (three electrode configuration and single cell 5 cm^2) of catalytic activity, with durability assessment and stability. There are clear indications of the origin of activity, mechanistic insights and novel findings by DFT on the role of tantalum oxide phase in contributing to the activity of iridium nanoparticles.

Response: We thank the Reviewer for his/her positive evaluation of our work. The reviewer gives an accurate summary of our work and brings forward constructive questions. We have modified our manuscript according to these valuable comments.

Major concern:

- The main motivation to introduce a supporting material is to reduce the amount of the most critical active material, like iridium in this case, increasing the activity per mass of substance. One of the requirements of a proper support is to be abundant, inexpensive and resistant. Within the advantages claimed for boride ceramics, only physical characteristics are mentioned. In my opinion, the authors must comment on the scarcity and expensiveness of tantalum and tantalum borides compared to PGM in order to really assess the suitability of this approach to reduce cost and improve feasibility of proton exchange membrane water electrolyzers. How does the cost of tantalum diboride compare to iridium dioxide? What about other potential diborides (Ti, Zr, etc.)? What about abundance and economic importance?

Response: Thanks for the kind and insightful comments and suggestions. We have compared the crustal abundance and price of Ir and Ta (**Table R1**). The abundance of Ta is five orders of magnitude higher than that of Ir, and Ta costs around 2% of the Ir price (\$367.5 for Ta vs. \$164662.0 for Ir per kilogram in 2023). In addition, the price of TaB₂ powder is \$1936.8 per kilogram, which is much cheaper than IrO₂ (\$261486.0 per kilogram), according to the selling prices of Adamas Reagent Co., Ltd. These comparisons confirm the suitability of TaB₂ supports to reduce cost of anode catalyst layer and improve feasibility of PEMWE. The corresponding discussion was involved in the revised manuscript (see **pages 6 and Supplementary Table 4**).

Table R1 The crust abundance and price of Ir, Ta, Zr and Ti.

Metal	Abundance of chemical elements in Earth's crust (ppm) ^a	Metal price (USD/kg, 2023) ^b
Ir	0.000003	164,662.0
Ta	2	367.5
Zr	190	28.6
Ti	5600	9.6

^a The abundance is obtained from: <https://environmentalchemistry.com/yogi/periodic/>.

^b The metal price is obtained from: <https://www.metal.com/price>.

Compared with TaB₂, other as-synthesized diborides are relatively unsuitable as catalyst supports for acidic OER. As shown in **Table R2 (or Supplementary Table 3)**, the TiB₂, ZrB₂, HfB₂, VB₂, CrB₂, MoB₂ and WB₂ show severe metal dissolution in acid, as well as the HfB₂, VB₂, NbB₂ and ReB₂ have low BET surface areas. Although Zr and Ti have higher crustal abundance and lower price than Ta (**Table R1**), their poor structural instability in acid makes them unsuitable as supporting material of acidic OER catalysts. Given the multi-advantages including large BET surface areas, great acid corrosion resistance and high conductivity, TaB₂ is selected as the most promising support in this work.

Table R2 The BET surface areas and leached metal contents of nine metal diborides.

Sample	BET Surface Area (m ² g ⁻¹)	Leached metal content (%) ^a
TiB ₂	45.8	14.6
ZrB ₂	43.3	52.6
HfB ₂	23.0	83.3
VB ₂	16.6	65.0
NbB ₂	27.0	0.1
TaB₂	54.6	0.04
MoB ₂	40.8	12.9
WB ₂	54.5	13.7
ReB ₂	8.6	19.1

^a The metal diborides were exposed to 0.5 M H₂SO₄ at 80 °C for 24 h and the leached metal content was obtained by ICP-OES test.

Minor concerns:

- Calculate the ECSA by normalizing by the mass of active phase (m² g_{Ir}⁻¹).

Response: We have calculated the ECSAs of IrO₂ and IrO₂@TaB₂ with different Ir content by normalizing the Ir mass (**Supplementary Table 6**). The Ir mass-normalized ECSA values become larger with increasing Ir content in IrO₂@TaB₂, and reach its maximum for the 16 wt% IrO₂@TaB₂

sample ($541 \text{ m}^2 \text{ g}_{\text{Ir}}^{-1}$).

- What are the units of the values included in Supplementary Table 1?

Response: The missing mass unit in Supplementary Table 1 is “g”. The mistake has been corrected in **Supplementary Table 1**.

- What conditions (more in particular potential window, scan rate, *etc.*) were used for the 5000 cycles (Supplementary Figure 9)?

Response: The 5000 CV cycles were performed in a potential window of 1.2-1.6 V vs. RHE at a scan rate of 100 mV/s. We have provided the information in caption of **Supplementary Figure 9**.

- Did you test lower content than 0.27 mg cm^{-2} of Pt at the cathode of the MEAs?

Response: We have provided the polarization curves of PEMWE using different content ($0.2\text{-}0.5 \text{ mg cm}^{-2}$) of Pt at the cathode (**Supplementary Figure 19**), while using $\text{IrO}_2\text{@TaB}_2$ anode under the same Ir loading of $0.15 \text{ mg}_{\text{Ir}} \text{ cm}^{-2}$. We consider 0.27 mg cm^{-2} as the optimized Pt loading at the cathode. When Pt content is increased to 0.5 mg cm^{-2} , the performance of PEMWE is not enhanced further. When Pt content is decreased to 0.2 mg cm^{-2} , the performance of PEMWE is decreased.

- In Figure 6e, the IrO_2 -based MEA presents very low stability. Taking into account that iridium catalysts exhibit good durability of up to thousands of hours at even larger current density (doi: 10.1016/j.apenergy.2020.115809), what is the reason for low stability?

Response: Response: Thanks for the reviewer’s good questions. The bad stability of PEMWE using bare IrO_2 anodes in **Figure 6e** is due to the low Ir loading of 0.2 mg cm^{-2} . The SEM image in **Supplementary Figure 20** exhibits the morphology and dispersion of IrO_2 catalyst on CCM. The IrO_2 aggregates only partially coat the electrode surfaces under the low Ir loading. In fact, the traditional CCMs in commercial PEMWE achieve a reasonable activity and stability using a much higher Ir loading of $2\text{-}4 \text{ mg}_{\text{Ir}}/\text{cm}^2$ to ensure sufficient in-plane conductivity and mechanical stability of catalyst layer (*Int. J. Hydrogen Energy* 2013, 38, 4901-4934). We further assess the performance of PEMWE containing a high Ir loading of $2 \text{ mg}/\text{cm}^2$ at anode layer (**Supplementary Figure 21**). As expected, while under the high Ir loading, the PEMWE using bare IrO_2 anodes exhibits excellent stability. Compared with bare IrO_2 , the $\text{IrO}_2\text{@TaB}_2$ with larger volume of material benefits to the stability of catalyst layer. The $\text{IrO}_2\text{@TaB}_2$ can form thicker catalyst layer relative to the bare IrO_2 , so that the former possesses sufficient in-plane conductivity and mechanical stability under a low Ir loading. The corresponding discussion was involved in the revised manuscript (see **pages 18**).

- Clearly indicate that the results depicted in figures 4c and 6d correspond to references of the supplementary information.

Response: The data in **Figures 4c** are obtained from cited **ref 1–8** in the Supporting Information. In addition, the data in **Figures 6d** are obtained from cited **ref 9–21** in the Supporting Information. We have provided the information in captions of **Figures 4c and 6d**.

- Include a reference for the DOE target in PEMWE in 2023.

Response: Thanks. We have provided ref 65 for the DOE target in PEMWE in 2023 in the revised manuscript.

- Please, carefully revise the manuscript for typo errors. Just to name some of them: captions of figure 1 are wrong; line 108 ‘angle of 60°C’; line 146 ‘unsuppoted’; lines 160 and 162 ‘suraces’; lines 524 and 525 ‘menberane’; etc. I have not indicated all the typos I found during reading, for practical reasons, so please revise it thoroughly.

Response: Thanks. The typos have been carefully checked and improved.

Reviewer #2 (Remarks to the Author):

The paper reports an interesting advance on the use of low Ir loading OER catalysts for PEM electrolysis. The subject is worthy of investigation as the high Ir loading is a major limitation to the mass deployment PEM electrolyzers. To achieve the target to cut the Ir loading down to 0.15 mg cm⁻² the authors propose the use of a heterojunction of IrO₂ with a TaB₂ support (Ta₂O₅ at the interface between the two component). The paper is well written and the conclusions are adequately supported by the deep characterization. I only have the following minor remarks:

Response: We appreciate reviewer #2 for his/her positive evaluation of our work. His/her comments lead to further improve the quality of our work. We have addressed them below.

1) What is the origin of the different stability of IrO₂ and IrO₂@TaB₂? Stability data are reported for 5 days. In this range the stability is excellent, but the stability the bare IrO₂ is extremely bad. Is this realistic? What is the origin of such dramatic difference? I wonder if the total amount of material present in the electrode plays a role here. The volume of the catalyst that contains the TaB₂ is at least twice the volume of the bare IrO₂. I invite the author to comment on this in the manuscript.

Response: Thanks for the reviewer’s good questions. The bad stability of PEMWE using bare IrO₂ anodes in **Figure 6e** is due to the low Ir loading of 0.2 mg cm⁻². The SEM image in **Supplementary Figure 20** exhibits the morphology and dispersion of IrO₂ catalyst on CCM. The

IrO₂ aggregates only partially coat the electrode surfaces under the low Ir loading. In fact, the traditional CCMs in commercial PEMWE achieve a reasonable activity and stability using a much higher Ir loading of 2-4 mg_{Ir}/cm² to ensure sufficient in-plane conductivity and mechanical stability of catalyst layer (*Int. J. Hydrogen Energy* 2013, 38, 4901-4934). We further assess the performance of PEMWE containing a high Ir loading of 2 mg/cm² at anode layer (**Supplementary Figure 21**). As expected, while under the high Ir loading, the PEMWE using bare IrO₂ anodes exhibits excellent stability.

We agree with this reviewer that the IrO₂@TaB₂ with larger volume of material benefits to the stability of catalyst layer. The IrO₂@TaB₂ can form thicker catalyst layer relative to the bare IrO₂, so that the former possesses sufficient in-plane conductivity and mechanical stability under a low Ir loading. The corresponding discussion was involved in the revised manuscript (see **pages 18**).

2) The English is generally good, but careful proofreading is needed to check for typos here and there.

Response: Thanks. The typos have been carefully checked and improved.

Reviewer #4 (Remarks to the Author):

The overall well-written manuscript by Chen, Zou and co-workers presents impressive performances for the oxygen evolution reaction via a carefully optimized IrO₂ catalyst supported on TaO_x/TaB₂. The comparison with the literature is convincing and one can expect the developed catalyst to become a new benchmark. The work on the stability and characterization of the catalyst is particularly valuable.

Response: We appreciate the reviewer for carefully reading our manuscript and making his/her insightful, critical, and constructive feedbacks.

The organization of the manuscript should be improved, as the choice of TaB₂ (as opposed to other borides) is justified after an already quite extensive presentation of this material.

Response: We agree with the reviewer's assessment. Accordingly, we have reorganized content in "*Fabrication and characterizations of nano-metal diborides*". In the revised manuscript, we first introduced the choice of TaB₂ as a promising support among as-synthesized nine metal diborides, followed by detailed characterizations of the TaB₂ sample.

Similarly, it is quite disappointing that the authors characterize and discuss in most detail the material with 16% IrO₂@TaB₂, but it is the 40% IrO₂@TaB₂ that is used for the actual device.

Response: We thank for the reviewer's valuable questions. Most of our characterizations and discussions focus on 16%IrO₂@TaB₂, because its highest activity among IrO₂@TaB₂ samples in three-electrode measurements. The three-electrode measurement is a well-established methodology for determining the mass activity and intrinsic activity of catalysts. OER activity through three-electrode measurement is typically evaluated in a very-low-current-density region (e.g. 10 mA cm⁻²), in which the electrochemical behaviour is dominated by reaction kinetics. The fundamental research focused on the most active sample (i.e. 16% IrO₂@TaB₂) will help revealing structure-activity relationship and catalytic mechanism.

However, the highly active catalysts evaluated in three-electrode measurement do not mean that they can yield high activity in PEMWE (*Mater. Chem. Front.* 2023,7, 1025-1045). In practical PEMWE tests where the operating current densities are mostly >1 A cm⁻², both reaction kinetics, electron transport and mass transport significantly affect the performance. Generally, the electrical conductivity of anode catalyst layer is required to be greater than 0.1 S·cm⁻¹ at PEMWEs (*Nano Res. Energy* 2022, 1, e9120032). Increasing the IrO₂ content supported on TaB₂ can achieve higher conductivity of catalyst. The electrical conductivity of 40%IrO₂@TaB₂ is up to 0.17 S cm⁻¹ (**Supplementary Figure 15**), which is similar to the conductivity of IrO₂ itself (0.18 S cm⁻¹). Hence, the 40%IrO₂@TaB₂ sample is employed as the optimal anode catalyst to integrate into PEMWE. The corresponding discussion was involved in the revised manuscript (see pages 16).

In addition, we have provided the structure and composition characterizations of the 40%IrO₂@TaB₂ sample by ICP, XRD, XPS (**Figure 2a-c**) and TEM (**Supplementary Figure 16**). The results showed that there is no essential difference between 40%IrO₂@TaB₂ and 16%IrO₂@TaB₂ sample, except for higher Ir loading of the former.

Given that the authors state themselves that the “TaB₂ support has no change to OER pathway of IrO₂”, the theoretical “justification” of the activity of the hybrid system is weak/farfetched: (a) experimentally it is mostly the stability that is enhanced and not the intrinsic activity and (b) the argument of “electron-rich IrO₂” is not very plausible under oxidative conditions: In contrast to state-of-the-art modelling, the authors did not consider grand-canonical DFT, in other words, the experimentally applied electrochemical potential is disregarded in the computations. Under oxidative conditions, one can reasonably expect the surface to be (partially) positively charged to reach the relevant potentials (see, for instance, 10.1039/D0CP00281J). In short, the computational aspect of the manuscript in its present form does, alas, not provide significant added value to the work of the authors.

Response: We thank the Reviewer for the comments and valuable suggestion.

Our response to the question (a): Experimentally, both the comparison results of activity normalized by electrode geometric area (**Figure 4a**), the activity normalized by ECSA (**Figure 4b**), and the iridium mass activity (**Figure 4c**) confirm that IrO₂@TaB₂ possesses significantly higher activity than the IrO₂. In addition, the PEMWEs using IrO₂@TaB₂ as anode electrocatalysts (1.9 V@2.5 A cm⁻²) also exhibits higher performance than that using IrO₂ as anode electrocatalyst (1.9 V@1.6 A cm⁻²), under the similar Ir loading (**Figure 6c**).

Our response to the question (b): For standard OER theoretical study on a given catalyst surface, the First principles DFT calculations were used to determine the free energy diagram and thereby the OER activity (*ChemCatChem* 2011, 3, 1159–1165), which neglects the effect of the applied potential. We agree with this reviewer that grand-canonical DFT is a more accurate method to model electrochemical processes of OER, with the ability to investigate the effect of applied potentials on the free energies of the reacting system. However, whether the standard DFT or grand-canonical DFT, the reasonable theoretical studies must be based on an accurate structural model of catalysts. For IrO₂@TaB₂ in this work, the uncertain hetero-interfaces of IrO₂ nanoparticles (without specific exposed facets) and amorphous TaO_x layer limit us to provide clear structural model for OER theoretical study. In view of the chemical and structural complexity of IrO₂@TaB₂ catalyst, we apologize for not being able to employ grand-canonical DFT for modeling electrochemical processes of OER.

Based on the above consideration, we calculate the electronic structure of IrO₂ to qualitatively discuss the effect of IrO₂/TaO_x electronic coupling on the chemisorption property and catalytic activity of IrO₂ (**Figure 5d-f**), rather than to quantitatively calculate the electrochemical processes of OER by establishing an unrealistic model structure. We very much look forward to employ the grand-canonical DFT calculations on a more suitable catalyst with well-defined surface in future research. The corresponding discussion was involved in the revised manuscript (see pages 15-16).

Detail:

The cited literature should include doi/10.1021/acscatal.0c04613.

Response: The literature has been cited in ref 38.

REVIEWERS' COMMENTS

Reviewer #1 (Remarks to the Author):

The authors have addressed the main important problems arisen during the reviewing process, with particular focus on the major concern. This reviewer still hesitates about the durability discussion for the reference iridium dioxide catalyst. The authors mainly attribute the low stability of iridium dioxide to the low loading of catalyst, showing that for higher loading the stability is much better. It is speculated that the better stability of the tantalum diboride supported IrO₂ catalyst is a result of a thicker layer with good electrical conductivity, but this is not yet in agreement with the state of the art stability for reference iridium dioxide even at such low loading close to 0.2 mg/cm². They also claim mechanical stability in the rebuttal letter, does this mean that low loading IrO₂ is detached from the electrode? In my opinion, a detailed discussion on this must be given before publication.

Reviewer #4 (Remarks to the Author):

The revision has clarified some aspects, but my major concern regarding the computational aspect has not been addressed in a satisfactory way: The authors write "The required overpotential to generate a current density of 10 mA cm⁻² (normalized over the electrode geometric area) for 16 wt% IrO₂@TaB₂ is 288 mV, lower than that of unsupported IrO₂ (307 mV). The Tafel slope of 16 wt% IrO₂@TaB₂ is 42.6 mV dec⁻¹ (Supplementary Figure 8), close to that of unsupported IrO₂ (45.1 mV dec⁻¹), indicating that the the TaB₂ support has no change to OER pathway of IrO₂"

And I agree with this statement: The over potential is slightly lower for the supported IrO₂ (but only 20 mV) and the Tafel slope is also very similar. In other words, from the performances at 10 mA, there is almost no difference between the two catalysts. It is only at higher current densities that they start to differ.

Suggesting that this difference would come from semi-conductor/conductor contacts, which equalise the Fermi-level between the two materials is far fetched, especially in an electrocatalytic device, where the Fermi-level of the operating material is imposed by the external potential anyway. It seems more likely that the anchoring of IrO₂ on the TaOx surface is responsible for slight morphological changes, maybe too subtle to be detected experimentally. I do agree with the authors that it would be very difficult to capture such a subtle effect on such a complex material.

In view of the overall speculative nature of the interpretation of the computational results, I would suggest to move them altogether to the SI and replace the current discussion with a vague hypothesis that the electronic level alignment (which the authors invoke) subtly changes the morphology during the synthesis of the material, which is, in some respect, also reflected in the increased stability.

This comment/suggestion does not diminish the value of the experimental work, though, which seems to be of high quality and should lead to quite some interest in the (large) community working on OER.

Reviewer's Comments and Our Responses:

Reviewer #1 (Remarks to the Author):

The authors have addressed the main important problems arisen during the reviewing process, with particular focus on the major concern. This reviewer still hesitates about the durability discussion for the reference iridium dioxide catalyst. The authors mainly attribute the low stability of iridium dioxide to the low loading of catalyst, showing that for higher loading the stability is much better. It is speculated that the better stability of the tantalum diboride supported IrO₂ catalyst is a result of a thicker layer with good electrical conductivity, but this is not yet in agreement with the state-of-the-art stability for reference iridium dioxide even at such low loading close to 0.2 mg/cm². They also claim mechanical stability in the rebuttal letter, does this mean that low loading IrO₂ is detached from the electrode? In my opinion, a detailed discussion on this must be given before publication.

Response: We thank the Reviewer for his/her positive evaluation of our work. In order to investigate degradation processes of PEMWE using 0.2 mg cm⁻² pure IrO₂ anode, we carry out chemical impedance spectroscopy (EIS) experiment after electrochemical testing, and fit the EIS Nyquist plots by equivalent electrical circuit (EEC) model (**Supplementary Figure 22**). The increase of the cell voltage was mainly due to large increases of the ohmic resistance and activation resistance. The ohmic resistance of the cell increases from 136 mΩ cm² to 190 mΩ cm², and the activation resistance increases from 42 mΩ cm² to 53 mΩ cm². The increased activation resistance also can be further supported by comparing the cyclic voltammograms before and after electrochemical testing of PEMWE using 0.2 mg cm⁻² pure IrO₂ anode. As shown in **Supplementary Figure 23**, the number of electrocatalytic active sites decreases after electrochemical testing. These results indicate that unfavorable microstructural evolutions (*e.g.*, agglomeration of catalyst particles, exfoliation of the catalyst layer) are obvious for low loading IrO₂ layer, leading to the reduction of electrocatalytic active sites and the increase of the ohmic resistance. We have provided above discussion in the Supplementary Information.

Reviewer #4 (Remarks to the Author):

The revision has clarified some aspects, but my major concern regarding the computational aspect has not been addressed in a satisfactory way: The authors write "The required overpotential to generate a current density of 10 mA cm⁻² (normalized over the electrode geometric area) for 16 wt% IrO₂@TaB₂ is 288 mV, lower than that of unsupported IrO₂ (307 mV). The Tafel slope of 16 wt% IrO₂@TaB₂ is 42.6 mV dec⁻¹ (Supplementary Figure 8), close to that of unsupported IrO₂

(45.1 mV dec⁻¹), indicating that the the TaB₂ support has no change to OER pathway of IrO₂".

And I agree with this statement: The over potential is slightly lower for the supported IrO₂ (but only 20 mV) and the Tafel slope is also very similar. In other words, from the performances at 10 mA, there is almost no difference between the two catalysts. It is only at higher current densities that they start to differ.

Suggesting that this difference would come from semi-conductor/conductor contacts, which equalise the Fermi-level between the two materials is farfetched, especially in an electrocatalytic device, where the Fermi-level of the operating material is imposed by the external potential anyway. It seems more likely that the anchoring of IrO₂ on the TaO_x surface is responsible for slight morphological changes, maybe too subtle to be detected experimentally. I do agree with the authors that it would be very difficult to capture such a subtle effect on such a complex material.

In view of the overall speculative nature of the interpretation of the computational results, I would suggest to move them altogether to the SI and replace the current discussion with a vague hypothesis that the electronic level alignment (which the authors invoke) subtly changes the morphology during the synthesis of the material, which is, in some respect, also reflected in the increased stability.

This comment/suggestion does not diminish the value of the experimental work, though, which seems to be of high quality and should lead to quite some interest in the (large) community working on OER.

Response: We thank the Reviewer for his/her positive evaluation of our work. According to the suggestion, we have moved the interpretation of computational results to the SI (**Supplementary Figure 15**). In addition, we have provided the following discussion in the revised manuscript (page 14-15).

“...After determining the catalytic mechanism, we sought to explain why IrO₂@TaB₂ possesses high activity for OER. Given that the crystal lattice and catalytic mechanism of IrO₂ have almost no change after loading on TaB₂, the enhanced performance of IrO₂@TaB₂ for OER is interpreted as follows. (i) The average particle sizes of IrO₂@TaB₂ and IrO₂ are estimated to be 1.5 nm and 1.8 nm (**Supplementary Figure 6**). Confinement of IrO₂ nanoparticles on the TaB₂ supports reduces IrO₂ size, which provides more active sites to improve OER activity. (ii) The interfacial electronic coupling in TaO_x/IrO₂ catalytic layer is responsible for the high intrinsic activity of IrO₂@TaB₂. A metal–semiconductor heterojunction is constructed between TaO_x and IrO₂ on TaB₂ surface, resulting in the formation of surface electric field and strong electronic interaction (**Supplementary Figure 15a**). The electrons will flow from conduction band of TaO_x to IrO₂, driven by the work function differences, leading to electron-rich IrO₂. The antibonding states of IrO₂ are more fully occupied by electrons in the IrO₂-TaO_x heterojunction, which lowers surface oxygen adsorption of IrO₂ and consequently boosts the OER activity (**Supplementary Figure 15b,c**)...”